# Robust assessment of asymmetric division in colon cancer cells

Domenico Caudo[1,2], Chiara Giannattasio[1,3], Simone Scalise[1,2], Valeria de Turris[1], Fabio Giavazzi[3], Giancarlo Ruocco[1,2], Giorgio Gosti[4], Giovanna Peruzzi[1], Mattia Miotto[1]*

[1]Center for Life Nano and Neuro Science, Istituto Italiano di Tecnologia, Rome, Italy; [2]Department of Physics, Sapienza University, Rome, Italy; [3]Department of Medical Biotechnology and Translational Medicine, University of Milan, Segrate, Italy; [4]Istituto di Scienze del Patrimonio Culturale, Consiglio Nazionale delle Ricerche, Montelibretti, Italy

## eLife Assessment

This study presents a **useful** method based on flow cytometry to study partitioning noise during cell division. The methods, data and analysis support the claims of the authors is **convincing**. This work will be of interest to cell biologists and biophysicists working on asymmetric partitioning during cell division.

*For correspondence: mattia.miotto@roma1.infn.it

**Abstract** Asymmetric partition of fate determinants during cell division is a hallmark of cell differentiation. Recent work suggested that such a mechanism is hijacked by cancer cells to increase both their phenotypic heterogeneity and plasticity and, in turn, their fitness. To quantify fluctuations in the partitioning of cellular elements, imaging-based approaches are used, whose accuracy is limited by the difficulty of detecting cell divisions. Our work addresses this gap, proposing a general method based on high-throughput flow cytometry measurements coupled with a theoretical framework. We applied our method to a panel of both normal and cancerous human colon cells, showing that different kinds of colon adenocarcinoma cells display very distinct extents of fluctuations in their cytoplasm partition, explained by an asymmetric division of their size. To test the accuracy of our population-level protocol, we directly measure the inherited fractions of cellular elements from extensive time lapses of live-cell laser scanning microscopy, finding excellent agreement across the cell types. Ultimately, our flow cytometry-based method promises to be accurate and easily applicable to a wide range of biological systems where the quantification of partition fluctuations would help account for the observed phenotypic heterogeneity and plasticity.

## Introduction

Asymmetric cell division refers to the mechanism by which a mother cell splits into two daughter cells with distinct cellular fates. As first shown by *Conklin, 1905*, this process is usually achieved by asymmetric inheritance of intrinsic determinants of cell fate, such as specific proteins or RNA, and plays a crucial role during the development of organisms, favoring cell differentiation and self-renewal (*Knoblich, 2001*; *Sunchu and Cabernard, 2020*).

Increasing evidence is vouching for a more ubiquitous presence of fluctuations in the partitioning of cellular elements. In fact, asymmetric segregation has been observed in non-differentiating cell populations as different as bacteria (*Mushnikov et al., 2019*, *Yang et al., 2015*), and tumor cells (*Buss et al., 2024*; *Chao et al., 2024*). In bacterial populations, differential segregation of cellular

components has been linked to antibiotic resistance (*Lindner et al., 2008*). Similarly, *Katajisto et al., 2015*, studying mitochondrial partition in stem-like mammalian epithelial cells, found that asymmetric segregation of older mitochondria enables cells to protect themselves from aging by preventing the accumulation of misfolded proteins. In addition, asymmetric centrosome partition has been observed in stem cells from human neuroblastoma and colorectal cancer, controlled by polarity factors and influenced by the segregation of subcellular vesicles during cell division (*Izumi and Kaneko, 2012*). To quantify the strength of fluctuations at division, the common route goes through fluorescence microscopy measurements of either fixed or live cells, where fluorescent dyes are used as a proxy for the cellular element of interest. The partition statistics are then estimated by looking at the fraction of fluorescence intensity inherited by the two daughter cells (*Dey-Guha et al., 2011*; *Dey-Guha et al., 2015*). This approach is, in principle, very informative and a lot of work has been done in recent years to write theoretical models of stochastic gene expression (*Beentjes et al., 2020*; *Jia and Grima, 2023*; *Dessalles et al., 2020*; *Thomas, 2019*) that can account for the coupling of noise sources and can even be extended to time-lapse microscopy data of single cell's growing dynamics *Jia and Grima, 2021*; however, relying on microscopy data implies the need for the (often manual) identification of hundreds of division events and in some cases even complex experimental setups and techniques where cells (usually bacteria) have to be followed for a great number of generations (*Jia and Grima, 2021*). These are highly time-consuming procedures that dampen the possibility of a wide characterization of the partition statistics of different cell types.

To cope with these limitations, we propose using high-throughput flow cytometry measurements to quantify fluctuations in the partitioning of cellular components in adherent cells. In particular, flow cytometry has already been applied to study asymmetric division in specific cellular systems: *Yang et al., 2015* sorted budding yeast cells to analyze the differential segregation of proteins between daughter cells, while Peruzzi and coworkers first applied multi-color flow cytometry to show that leukemia cells divide mitochondria and membrane proteins asymmetrically (*Peruzzi et al., 2021*). Here, we generalized the protocol to adherent cell types and probed its accuracy by comparing the measured fluctuations to those observed with extensive time-lapse microscopy experiments on a panel of normal and cancerous epithelial cells.

Our work shows that (1) there is an exact analytical expression linking the inherited cellular component distribution dynamic to the specifics of the partition process. Based on this result, we (2) proposed an experimental protocol to tag with fluorescent dyes cellular components and measure the dynamics of fluorescent distribution, whose analysis allows accurate estimates of the degree of asymmetry in the partition process. Applying our method to a panel of normal and cancerous human colon cells, we found that (3) different lines of colon adenocarcinoma display very distinct extents of fluctuations in cytoplasm partition, reflecting in (4) different degrees of asymmetric division of their size.

## Results

### Modeling cell population dynamics in the presence of partitioning noise

In this section, we propose a theoretical framework that is able to describe the evolution of the fluorescence intensity of a cell population stained with a live fluorescent marker, by uncovering its dependence on the underlying partitioning process to which single cells are subjected (see *Figure 1a-c*). The rationale is that, if cells follow a common division rule, knowing the shape of the mother distribution, it should be able to determine the daughters' one. By iterating this computation, one can trace the whole proliferation. The specificity of our approach lies in its specific design for live fluorescent markers. The selected component is tagged through a staining procedure with fluorophores that persist in the cell and its daughter cells, as long as the targeted protein is not degraded.

Monitoring the time evolution of the abundance of specific cellular elements over multiple divisions in single-cell experiments, it is observed (*Soltani et al., 2016*) that the counts of a component have a qualitatively similar dynamic to those shown in *Figure 1b*. Indeed, variations in the number of components are determined by (1) the production and degradation processes along the duration of the cell growth phase; and (2) non-identical apportioning of the components between the two daughter cells. Here, we will assume that the number of components varies only due to partition events and look for a quantitative description of the evolution of the element distribution. By neglecting variability in division times and the intercurrent production and degradation of components, we are reducing

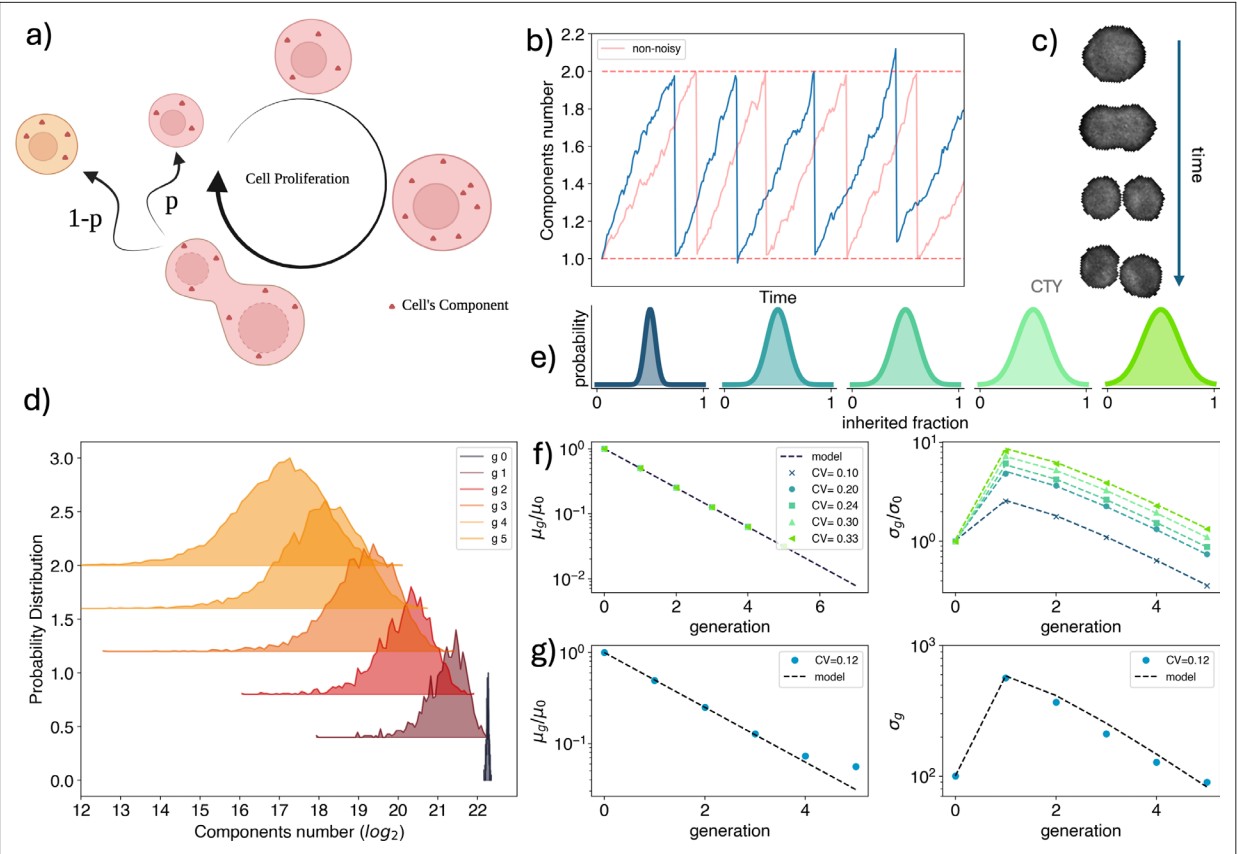

**Figure 1.** Modeling partitioning noise at cell division. (**a**) Schematic representation of the growth and division process of a mother cell. The cell undergoes an initial phase of duplication, where its internal elements are multiplied, followed by a division phase, where these elements are partitioned between the two daughter cells. One daughter inherits a fraction $p$ of the mother's elements, while the other receives the remaining fraction $q = 1 - p$. (**b**) (Blue) Idealized behavior of the components number of a cell element over time, considering three different noise terms: fluctuations in compound counts during growth, uncertainty in the timing of division, and noise in the compound's partition fraction. (Red) Same description, but considering only production and degradation processes. (**c**) Microscopy images of a single-cell division event for an HCT116 cell, whose cytoplasm is stained with Celltrace Yellow. (**d**) Time evolution of the distribution of the components' number of a cellular element for a population of cells subjected only to partitioning noise. The distribution used for the simulation is a Gaussian distribution with mean $\mu = \frac{1}{2}$ and standard deviation $\sigma = 0.07$. (**e**) Examples of partition distributions $\Pi$, with increasing coefficient of variation. (**f**) Mean ($\mu_g$) and variance ($\sigma_g$) of the number of components as a function of the generations, $g$, for the proliferation of a population subject to different partition noise distributions. Different colors correspond to distributions with varying coefficients of variation, as represented in the top panels. The dashed line represents the theoretical behavior obtained from the model. Dots are colored according to the distributions shown in panel (**e**). (**g**) Behavior of the mean (left) and standard deviation (right) of the simulated dynamics compared to the expected theoretical behavior for a proliferating population assuming a sizer division strategy (see Appendix).

The online version of this article includes the following figure supplement(s) for figure 1:

**Figure supplement 1.** Mean (left) and variance (right) of the distribution of fluorescence intensity as a function of the generations for simulations with cell-cycle length variability.

**Figure supplement 2.** Stability to coupled noises.

**Figure supplement 3.** Fit of the coupled noise simulations.

the dynamics to a progressive dilution. The stability of our model with respect to these assumptions will be studied in the following section and more in detail in the Appendix. Note that it is possible to model the complete growth and division process (see, for instance, *Miotto et al., 2023*; *Scalise et al., 2024*; *Jia and Grima, 2023*; *Thomas, 2018*), but it is more difficult to separate the different sources of noise. Here, we start by considering a population of cells characterized by a certain initial distribution, $P(m)_0$, of element m. Each cell $i$ of the population divides into two daughter cells that we can label unambiguously as $2i$ and $2i + 1$, which inherit, respectively, $m_{2i}$ and $m_{2i+1}$ components. The process, exactly at division, must conserve the total number of elements, that is, $m_i = m_{2i} + m_{2i+1}$, and we call the partitioning fraction the quantity $f = \frac{m_{2i}}{m_i}$. To model noise in partitioning, we assume that at each

division, a random value for $f$ is extracted from a probability distribution function, $\Pi(f)$. Given these assumptions, we can write the probability distribution of the number of components of a daughter cell $m$ after $g$ divisions from the mother $M$ as:

$$P_g(m) = \left(\frac{1}{2^{g-1}}\right) \sum_{k=0}^{2^{g-1}} \int dM P(m|M) P_{g-1}^k(M),$$ (1)

where $P_{g-1}^k(M)$ are the components subdistributions present at generation $g-1$. After $g$ generations, the cell lineage generated from a single-cell forms a lineage tree composed of $n_g = 2^g$ cells. Without losing generality, the division probability can be expressed as:

$$P(m|M) = \int \delta(m - fM)\Pi(f)df.$$ (2)

Therefore, we can write the probability distribution, $P(m)$, of the daughter cells inheriting a fraction $f$ of the mother elements:

$$P(m) = \int df dM \delta(m - fM)\Pi(f)P(M).$$ (3)

As is often the case, it is convenient to characterize the distributions in terms of their moments (***Miotto et al., 2024***). We identify with $m_{2i}$ ($m_{2i+1}$) the subpopulation of cells inheriting a fraction $f$ $(1-f)$. In particular, the mean of the daughter cell component distribution can be expressed, for $2i$, as:

$$\mu_{2i} = E[m_{2i}] = \int P(m_{2i})m_{2i}\, dm_{2i}$$

$$= \int df\, dm_i\, dm_{2i}\, m_{2i}\delta(m_{2i} - fm_i)\Pi(f)P(m_i)$$ (4)

$$= E[f]E[m_i] = \mu_f \mu_i,$$

where $E[\cdot]$ stands for the average. Note that every computation can be symmetrically done for the siblings subpopulation $2i + 1$. Thus, we have that after a single generation, the mean number of inherited elements of the daughter cells depends on the asymmetry of the $\Pi(f)$ and on the mean of the $P(m_i)$.

In ***Figure 1d***, we show the evolution of the components number distribution at different generations. The $\Pi(f)$ chosen for the simulation is a Gaussian distribution with mean $\mu = \frac{1}{2}$ and standard deviation $\sigma = 0.7$.

The total distribution is the sum of $2^g$ subpopulations, which is derived from the division process. For example, if we assume $\Pi(f) = \frac{1}{2}(\delta(f) + \delta(1 - f))$, after one division, we would have a population distributed around $f\mu$ and one at $(1 - f)\mu$. After two divisions, the populations would grow to four subpopulations centered in $f^2\mu$, $f(1-f)\mu$, $(1-f)f\mu$, $(1-f)^2\mu$, and so on.

To compute the mean relative to an entire generation, we need to sum over all the different subpopulations:

$$\mu_g = \frac{1}{2^g} \sum_{k=1}^{2^g} \mu_g^k,$$ (5)

where $\mu_g^k$ is the mean of the $k$th subpopulation at generation $g$ and $2^g$ is the total number of subpopulations for that generation. ***Equation 5*** can be simplified into the form (see Appendix for the full computation):

$$\mu_g = \left(\frac{1}{2}\right)^g \mu_0$$ (6)

where $\mu_0$ is the mean of the initial population. It is relevant to notice that independently of the characteristic of $\Pi(f)$, the mean always halves at each generation. This result is simply understood considering that for each cell that inherits a fraction $f'$ of the compound, its sibling inherits a fraction $1 - f'$.

Hence, at every division, the compound count is diluted by an average factor of 2 in the population, and partition distributions with different properties lead to the same behavior of $\mu_g$ (see **Figure 1f**). Therefore, no relevant insights on the $\Pi(f)$ can be obtained from the first moment. We thus moved to consider the second moment, that is, the variance. Similarly to what is done for the mean, the variance of the inherited fraction of elements can be expressed as:

$$\sigma_{2i}^2 = E[m_{2i}^2] - E[m_{2i}]^2$$
$$= \int df \, dm_i \, dm_{2i} \, m_{2i}^2 \delta(m_{2i} - fm_i)\Pi(f)P(m_i) - \mu_f^2 \mu_i^2 \tag{7}$$
$$= E[f^2]E[m_i^2] - \mu_f^2 \mu_i^2 = E[f^2]\sigma_i^2 + \mu_i^2 \sigma_f^2,$$

and in turn, the variance of a generation, starting from the consideration of the variance of a mixture of distributions, can be written as (see Appendix for details):

$$\sigma_g^2 = \frac{1}{2^g} \sum_{k=1}^{2^g} \left( \sigma_{g,k}^2 + \mu_{g,k}^2 \right) - \mu_g^2 = A_g - \mu_g^2, \tag{8}$$

with

$$A_g = \frac{1}{2^g} \sum_{k=1}^{2^g} \left( \sigma_{g,k}^2 + \mu_{g,k}^2 \right). \tag{9}$$

It is possible to obtain a recursive equation for $A_g$ (see Appendix), $A_g = A_{g-1}E[f^2] = A_0E[f^2]^g$ which leads to a compact form for **Equation 8**:

$$\sigma_g^2 = \mu_0^2(E[f^2]^g - (1/2)^{2g}) + \sigma_0^2 E[f^2]^g, \tag{10}$$

where $E[f^2] = \sigma_f^2 + \mu_f^2 = \sigma_f^2 + 1/4$.

Finally, with **Equation 10**, we can link the variance of the components distribution for the entire cell population at generation $g$ to the properties of $\Pi(f)$. It is interesting to highlight that there is no dependence on the bias asymmetry of the process. The distribution $\Pi(f)$ is necessarily symmetric, as daughter cells are indistinguishable. Therefore, by bias, we refer to a situation where the most probable outcome of the division process is asymmetric. To this order, the only relevant feature shaping the dynamics is the extent of the fluctuations, regardless of their origin. Indeed, as shown in **Figure 1e**, the same $\Pi(f)$, but with a different variance, shows a different behavior for $\sigma_g^2$.

Our analytical results show that the dynamical behavior of the population variance depends non-trivially on the second moment of the underlying partitioning distribution. The next section will be devoted to demonstrating how this condition can be exploited in experiments to measure the partition noise component and to evaluate the stability of our assumptions with respect to the concurrence of the other noise sources (see also **Figure 1g**).

## Measuring fluctuations via flow cytometry

As shown by **Peruzzi et al., 2021** for leukemia cells growing in suspension, it is possible to use flow cytometry to follow the evolution of properly marked cellular elements in time. Here, we generalized the protocol considering a panel of colon cell lines, that is, Caco2, a human epithelial colorectal adenocarcinoma cell line; HCT116, a human colorectal carcinoma cell line; and CCD18Co, a normal human colon fibroblast cell line. To measure the degree of asymmetric division, we marked cells' cytoplasm via a fluorescent dye and followed its dilution through successive generations. As explained in detail in Materials and methods, the experimental procedure we used is based on the following main steps: staining of the selected compound, sorting of the initial population, plating, and acquisition of the distinct populations in the following days. The sorting allows us to obtain an initial population with a narrow peak at a specific fluorescence intensity, and it plays a dual role (an extended discussion can be found in the Appendix): it indirectly induces cell synchronization and enables a clearer reconstruction of the coexisting generations.

To achieve this, we used fluorescence-activated cell sorting (FACS) to isolate a specific cell population based on cell morphology and fluorescence intensity. The sorted population is divided into

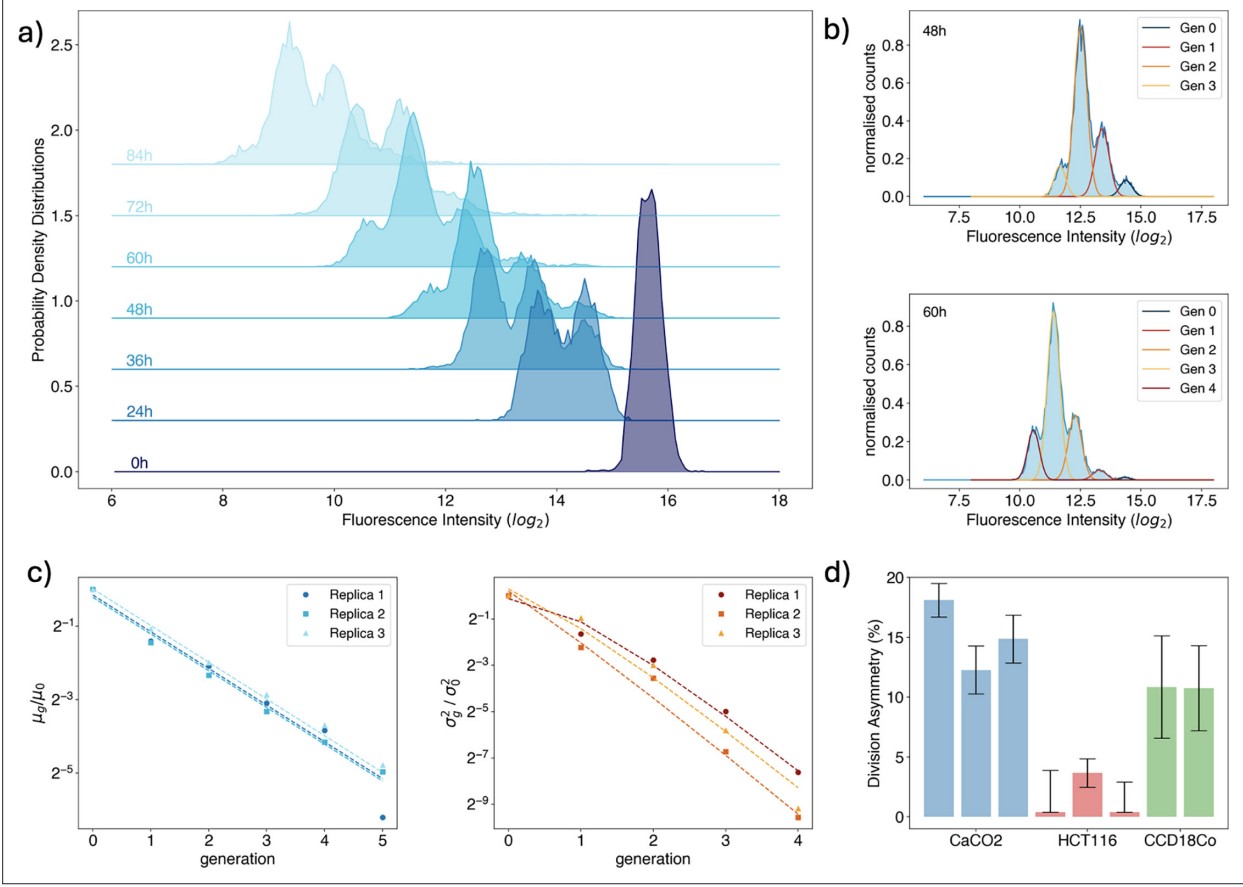

**Figure 2.** Quantification of partitioning noise via population-level measurements. (**a**) Time evolution of CellTrace-Violet fluorescence intensity distribution measured in a flow cytometry time course experiment for a population of HCT116 cells. Time progresses from the darkest shade of blue to the lightest, spanning from [0, 84] hr (bottom line to top). (**b**) Snapshots of the evolution of the distribution of CellTrace-Violet fluorescence intensity measured in a flow cytometry time course experiment for a population of HCT116 cells. Experimental data are represented by the light blue histogram, while the best-fit Gaussian mixture model is displayed as lines, with different colors representing different generations. (**c**) Mean (left) and variance (right) of the intensity of the fluorescent markers as a function of generations, normalized to the initial population values. Each replica of the experiment is identified by a different color and a point marker's shape. The experiments are conducted on Caco2 cells, and the points correspond to the mean values for each generation. Dashed lines represent the best fit according to *Equations 6 and 10*, respectively. (**d**) Division asymmetry of the Π(*f*) obtained by fitting *Equation 10* to data for all experiments and cell lines. The division asymmetry is measured via the percentage of the coefficient of variation.

The online version of this article includes the following figure supplement(s) for figure 2:

**Figure supplement 1.** Gate strategy for isolation of CTV-positive HCT-116 cells.

**Figure supplement 2.** Components distribution at varying coefficient of variations (CVs) of initial components and partitioning distributions.

**Figure supplement 3.** Resume of experimental data and fit.

**Figure supplement 4.** Replica 1 of Caco2 cells experiments.

**Figure supplement 5.** Replica 2 of Caco2 cells experiments.

**Figure supplement 6.** Replica 3 of Caco2 cells experiments.

**Figure supplement 7.** Replica 1 of CCD18Co cells experiments.

**Figure supplement 8.** Replica 2 of CCD18Co cells experiments.

**Figure supplement 9.** Replica 1 of HCT-116 cells experiments.

**Figure supplement 10.** Replica 2 of HCT-116 cells experiments.

**Figure supplement 11.** Replica 3 of HCT-116 cells experiments.

distinct wells, one per acquisition, in equal numbers and kept under identical growth conditions to mitigate inoculum-dependent variability among samples (*Enrico Bena et al., 2021*). *Figure 2a* shows an example of the time course dynamics obtained with HCT116 cells (see Methods for details). Each distribution represents a different acquisition of a whole well at increasing times from plating (0–84 hr from the bottom to the top). It is possible to neatly observe the succession of peaks associated with different generations that succeed one another and the average decrease of the fluorescence intensity. Note that different generations can coexist at the same time point. One can spot the flow of cells toward higher generations and the behavior of the peak heights, which grow over time, reach dominance, and then decay. The shift on the *y*-axis allows us to follow the evolutionary dynamics while being able to see the overlapping of the same generation at different time points. As in each measurement, the distribution of fluorescent intensity, $I$, is a mixture of distributions corresponding to different generations:

$$p(I) = \sum_g \pi_g P(I_g).$$

(11)

To obtain the necessary information on the properties of each generation, we performed a fitting of the data via a Gaussian mixture model (GMM) combined with an expectation–maximization algorithm (details are explained in Materials and methods). Indeed, assuming that each generation is log-normally distributed with mean $\mu_g$ and variance $\sigma_g^2$ of the corresponding Gaussian distribution, *Equation 11* can be rewritten as:

$$p(log_2(I)) = \sum_g \pi_g \mathcal{N}(log_2(I_g)|\mu_g, \sigma_g).$$

An example of the outcome of the GMM algorithm is shown in *Figure 2b* for two acquisitions (48 and 60 hr from plating). Blue-shaded distributions represent experimental data, while the best solution of the Gaussian mixture is shown with a solid line. Each generation's fit is identified with a different color. Upon an overnight, we can identify generations 0, 1, and 2 in both acquisitions. One can see that their relative fractions changed: generation 3 became dominant overnight, and generation 4 appeared.

*Figure 2c* shows the behaviors of $\mu_g$ and $\sigma_g$ as functions of the generation number $g$ for three independent replicas of the time course experiments with Caco2 cells. Dashed lines represent the best fit of the data against *Equation 5* for the mean and *Equation 10*, which instead yields the variance $\sigma_f$ of the partition distribution probability, $\Pi(f)$. The outcomes of the fitting procedure on the time course dynamic for all the replicas of the three studied cell lines are reported in *Figure 2d*. To evaluate the degree of asymmetry, we have defined the *division asymmetry* as the percentage of the coefficient of variation of the obtained partition distribution (CV: $\frac{\sigma}{\mu}$). In particular, Caco2 and CCD18Co cells show a higher level of asymmetry with respect to HCT116. As already stated in the description of the theoretical model, we consider only partitioning noise and neglect other potential sources: (1) production and degradation processes, and (2) variability in cell cycle length. Since we use non-endogenous live fluorescent dyes, production processes are excluded by definition, while degradation results in a negligible time-dependent decrease in mean fluorescence intensity as the used dyes are optimized to maintain stable fluorescence levels for the duration of the experiment. Indeed, by looking at the data in *Figure 2c*, the values of the measured $\mu_g$ seem to only fluctuate around the theoretical line, but with the same slope, indicating that degradation can be neglected without major consequences. The effects of correlations between the inherited fractions of cellular components and the fluctuations in the duration of the cell cycle are, in principle, more difficult to disentangle. In this respect, it is widely known that cells adjust their growth/division strategy to mitigate size fluctuations at division (*Thomas, 2018*; *Miotto et al., 2024*). According to the 'sizer' model, they divide upon reaching a certain size so that smaller cells, that is, those that inherit a smaller fraction of their mother's volume, will have longer division times and vice versa. This results in fluctuations in the cell division times that may be coupled with partitioning noise.

To assess the robustness of our model under such conditions, we performed simulations assuming that cells follow a sizer-like division strategy, while the marked component partition perfectly matches the cell size partition (see Appendix for more details). This scenario maximizes the coupling between the two noise sources. As can be observed in *Figure 1g*, the behaviors of the mean value $\mu_g$ and the

variance $\sigma_g$ deviate from the case in which division time is uncoupled with partition noise. However, the values obtained are still within the estimated experimental error (see *Figure 1—figure supplement 3*). It is worth noticing that the presence/absence of the increase in the mean, $\mu_g$, for newer generations may highlight the presence of the coupling between the two noise sources.

## Validation of the method via live fluorescence microscopy

The flow cytometry-based protocol combined with the developed theoretical model provides insights into the properties of the cell partition function (see *Figure 2*). To validate our approach, we performed extensive live time-lapse microscopy experiments to measure the partition fraction of a labeled compound during proliferation, providing a direct measure of the $\Pi$ ($f$). Specifically, we aimed to follow a growing cell colony over time, detect divisions, evaluate the fluorescence intensity of mother and daughter cells, and calculate the inherited fraction. The detailed experimental procedure is described in the Materials and methods. Briefly, cells are stained for the cytoplasm and are plated at low density (approximately $10^3$ cells/well) on IBIDI cell imaging chambers (μ-Slide 4 and 8 wells).

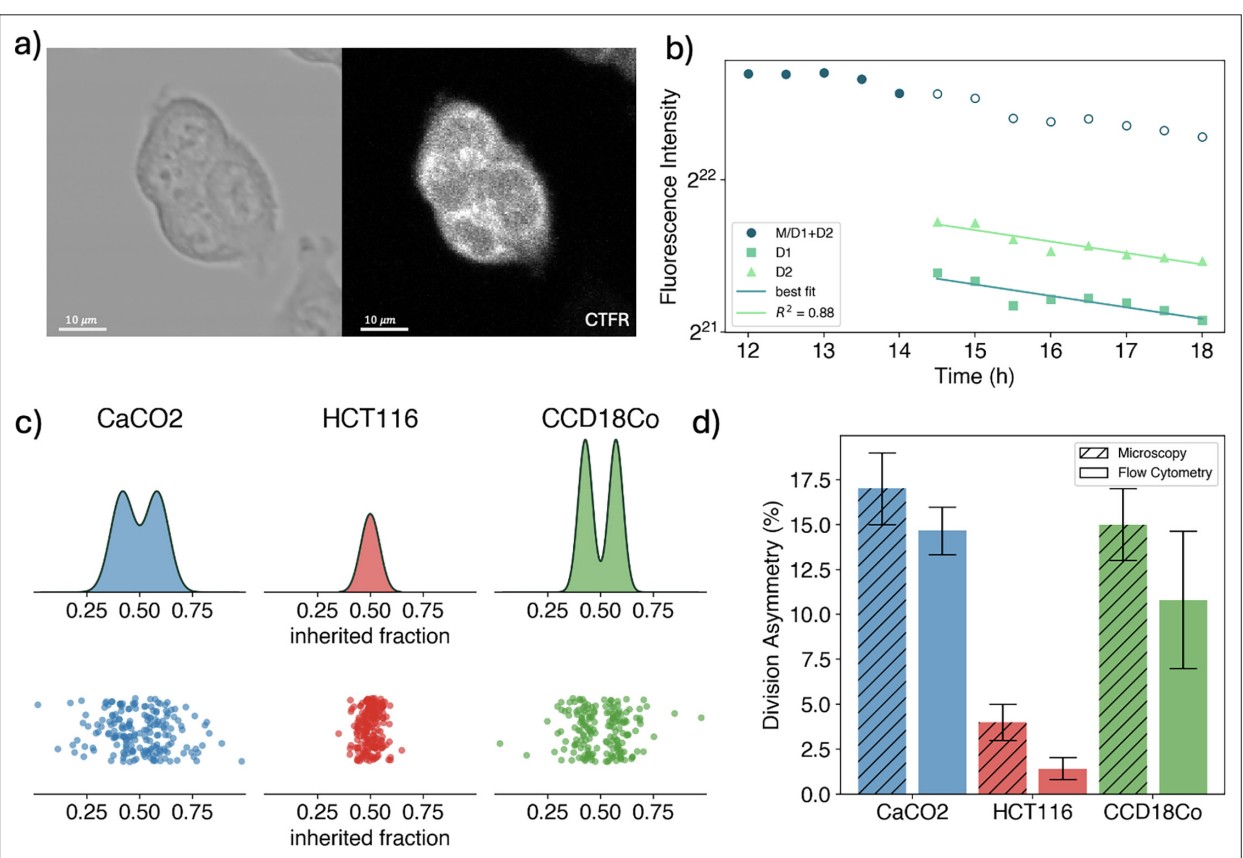

**Figure 3.** Quantification of partitioning noise via single-cell measurements. (**a**) Example of a recorded cell colony of HCT116 cells in brightfield (left) and on CTFR fluorescence (right). (**b**) Cell cytoplasm fluorescence intensity as a function of time for a cell before and after division. Dark green circles correspond to the fluorescence intensity of the mother cell up to the division frame, and then to the sum of the daughters' fluorescence. Lighter green triangles and squares represent the fluorescence intensity of the daughter cells. Solid lines are the linear fit of the points. The intercepts of the linear fit are used to compute the fraction of tagged components inherited by the daughter cells. The time is counted from the start of the experiment. (**c**) (bottom) Strip plot of the distribution of the inherited fraction of cytoplasm for the different cell lines. The points are randomly spread on the *y*-axis to avoid overlay. (Top) Fit of the inherited fraction distribution with the sum of two Gaussians with a mean symmetric to 1/2. (**d**) Comparison of division asymmetry obtained with time-lapse fluorescent microscopy measures (striped bars) with the ones obtained from flow cytometry experiments (plain bars). The flow cytometry bars are obtained as the mean over the multiple conducted experiments.

The online version of this article includes the following figure supplement(s) for figure 3:

**Figure supplement 1.** Simulated noisy dynamics of the fluorescence intensity.

**Figure supplement 2.** Comparison between the standard deviation of the partitioning distribution for the noisy and noise-free dynamics, shown in red and blue, respectively.

After 24 hr (sufficient time for complete cell adhesion to the slide), the wells are washed, the media is renewed, and live imaging acquisition time-lapse is started. Multiple fields are followed in each well, and for each field, we acquire a z-stack brightfield and fluorescence image (a single plane is shown in *Figure 3a*) at constant intervals (20 min) for 3 days. The fields are manually selected at the beginning of the acquisition based on cell density and fluorescence intensity. The recorded time lapses have been manually analyzed to identify divisions. A sample outcome of the analysis procedure is shown in *Figure 3b*. The total fluorescence of the cells is displayed as a function of time, before and after the division. The mother cell fluorescence decays exponentially over time, and at division, it halves due to components partitioning between the two daughters. In this specific case, we can observe that the fraction of cytoplasm inherited by the two cells is not equal, indicating an asymmetrical division. To compute the partition fraction $f$, we fit the fluorescence intensity of the daughters' cells versus time with $log(I_{2i}(t)) = mt + q_{2i}$ and $log(I_{21+1}(t)) = mt + q_{21+1}$, where $I(t)$ is the fluorescence intensity at time $t$, $m$ accounts for the decaying process and $2i$ and $2i + 1$, respectively, refer to the two daughter cells. We constrain the slope to be the same between the two cells. The fraction of inherited fluorescence, $f$, is obtained as:

$$f_{2i} = \frac{e^{q_{2i}}}{e^{q_{2i}} + e^{q_{2i+1}}} \quad \text{and} \quad f_{2i+1} = 1 - f_{2i}. \tag{12}$$

To ensure the highest possible accuracy, we (1) measured the total fluorescence of the mother cell and those of the two daughter cells for at least 2 hr before and after the division event to increase the determination of the fluorescence splitting for each detected mitosis. Moreover, (2) we removed all dynamics that showed an absolute Pearson correlation value between luminosity and pixel size higher than 0.9 as a function of time (see Appendix). In fact, single cell's pixel size, which corresponds to the cell size projection on the focal plane, is found to shrink before mitosis and to enlarge after division. Fluorescent intensity is instead proportional to the number of stained cytoplasmic proteins, which are expected to remain constant (except for a constant decay of the fluorescence). Therefore, a high correlation between pixel size and fluorescence intensity indicates a high noise-to-signal ratio. *Figure 3c* (bottom) displays the strip plot of the obtained partition fraction, $f$, for Caco2, HCT116, and CCD18Co cells, randomly spread on the y-axis, and their corresponding fitted $\Pi(f)$ distributions (top). Note that via microscopy imaging, it is possible to measure the whole partition distribution and not just its moments; however, hundreds of events are required to get a reliable estimate. Here, we characterize it by fitting the data with a double Gaussian distribution of the form:

$$\mathcal{N}(1/2, \sigma) = \frac{\mathcal{N}(f, \sigma') + \mathcal{N}(1 - f, \sigma')}{2}$$

that allows for the measure of both $\langle f \rangle$ and $\sigma_f^2 = \sigma'^2 + f^2 - f + 1/4$.

The fitted distributions clearly show how HCT116 cells are the most symmetric, with only one central and narrow peak, while for CCD18Co and Caco2 cells, two asymmetric peaks are visible, associated with higher fluctuations. In *Figure 3d*, we compare the outcomes of the flow cytometry versus microscopy measurements. For all three cell lines, the found degrees of asymmetry are statistically consistent between the two approaches.

## Size division bias accounts for cytoplasmic fluctuations

In the above sections, we have proposed and validated a method to evaluate fluctuations in the division of adherent cells. The model makes no assumptions about the shape of the partitioning distribution and, in its general form, accounts only for the effect of distribution variance in shaping the population dynamics. Indeed, we cannot yet distinguish between divisions which are biased, defined by a partitioning distribution with peak values different from 1/2 (as shown in *Figure 3c* for Caco2 and CCD18Co cells) or symmetric ones (as shown in *Figure 3c* for HCT-116 cells) which though have higher fluctuations. Also, apart from measuring it, we want to propose a qualitative interpretation of the obtained level of asymmetry. We start by recalling that our experimental protocol uses dyes that bind a-specifically to cytoplasmic amines. Thus, (1) it is expected to be uniformly distributed in the cellular cytoplasmic space and (2) the number of labeled cytoplasmic components can be considered large. With this hypothesis, the least complex model one can assume is a binomial one with parameter $p$, measuring the bias in the process. Via *Equation 10*, we got a direct link between the measurable

variance of the population and the second moment of the underlying partition probability distribution, $\Pi(f)$. Henceforth, we sought a relationship between the second moment of $\Pi(f)$ and the parameters of the binomial distribution. To begin with, we can write the partition distribution for the fraction of inherited component, assuming a level of asymmetry $p$ ($q = 1 - p$), as:

$$\Pi(f) = \frac{1}{2}(\Pi(f)^{(p)} + \Pi(f)^{(q)}). \tag{13}$$

As we have already observed in the general model, due to symmetry, $\langle f \rangle = 1/2$ for any form of the $\Pi(f)$. No information on the system can be obtained by looking at the first moment.

For the second moment, we note that the variance $\sigma_f^{2\,p}$ of a single branch of the $\Pi(f)$ is:

$$\begin{aligned} \sigma_f^{2\,(p)} &= \langle f^2 \rangle^{(p)} - \langle f \rangle^{2\,(p)} \\ &= \int dN_i\, \sigma_{f|N_i}^{2\,(p)}\, P(N_i)' \end{aligned} \tag{14}$$

where $P(N_i)$ is the probability distribution of the mother's components number at the moment of division, while $\sigma_{f|N_i}^{2\,(p)}$, is the variance of the $\Pi(f)^p$ given the number of internal component $N_i$ in the mother cell.

Recalling that $m_{2i}$ is the number of inherited intracellular components of one of the two daughter cells, the partitioning fraction $f$ can be expressed as $f = m_{2i}/N_i$, which leads to:

$$\begin{aligned} \sigma_{f|N_i}^{2\,(p)} &= \langle f^2 \rangle_{N_i}^{(p)} - \langle f \rangle_{N_i}^{2\,(p)} \\ &= \frac{1}{N_i^2}\, \sigma_{m_i}^{2\,(p)} = \frac{1}{N_i^2}\, N_i p q = \frac{pq}{N_i}. \end{aligned} \tag{15}$$

Therefore, by substituting it into *Equation 14* we obtain:

$$\sigma_f^{2\,(p)} = pq \int dN_i\, \frac{1}{N_i} P(N_i), \tag{16}$$

which explicitly depends on mother's component probability distribution, $P(N_i)$, and coherently returns $\sigma_{m_i}^{2\,(p)} = \frac{pq}{N}$ for $P(N_i) = \delta(N_i - N)$.

An equivalent computation can be done for the symmetric branch ($\sigma_{m_i}^{2\,(p)}$) and therefore the variance of the full distribution is given by:

$$\begin{aligned} \sigma_f^2 &= \langle f^2 \rangle - \langle f \rangle^2 \\ &= \frac{1}{2}\left( \langle f^2 \rangle^{(p)} - \langle f^2 \rangle^{(q)} \right) - \frac{1}{4} \\ &= pq \int dN_i\, \frac{1}{N_i} P(N_i) + \frac{p^2 + q^2}{2} - \frac{1}{4} \end{aligned} \tag{17}$$

Note that this relationship depends on the integral:

$$\Sigma(N) = \int dN_i\, \frac{1}{N_i} P(N_i)$$

which requires the knowledge of the mother element distribution $P(N_i)$ to be computed. To get an idea of its behavior, we observe that assuming a population of cells with a fixed number of elements before division, that is, $P(N_i) = \delta(N - N_i)$, the integral simply goes as $1/N$; in a more realistic scenario, the elements can be considered log-normally distributed, since they are the product of multiple division processes, hence:

$$P(N_i) = \frac{1}{N_i \sqrt{2\pi \sigma_{N_i}^2}} \exp\left( \frac{-(\ln(N_i) - \mu)^2}{2\sigma_{N_i}^2} \right).$$

In this case, one gets the following expression for $\sigma_f^2$:

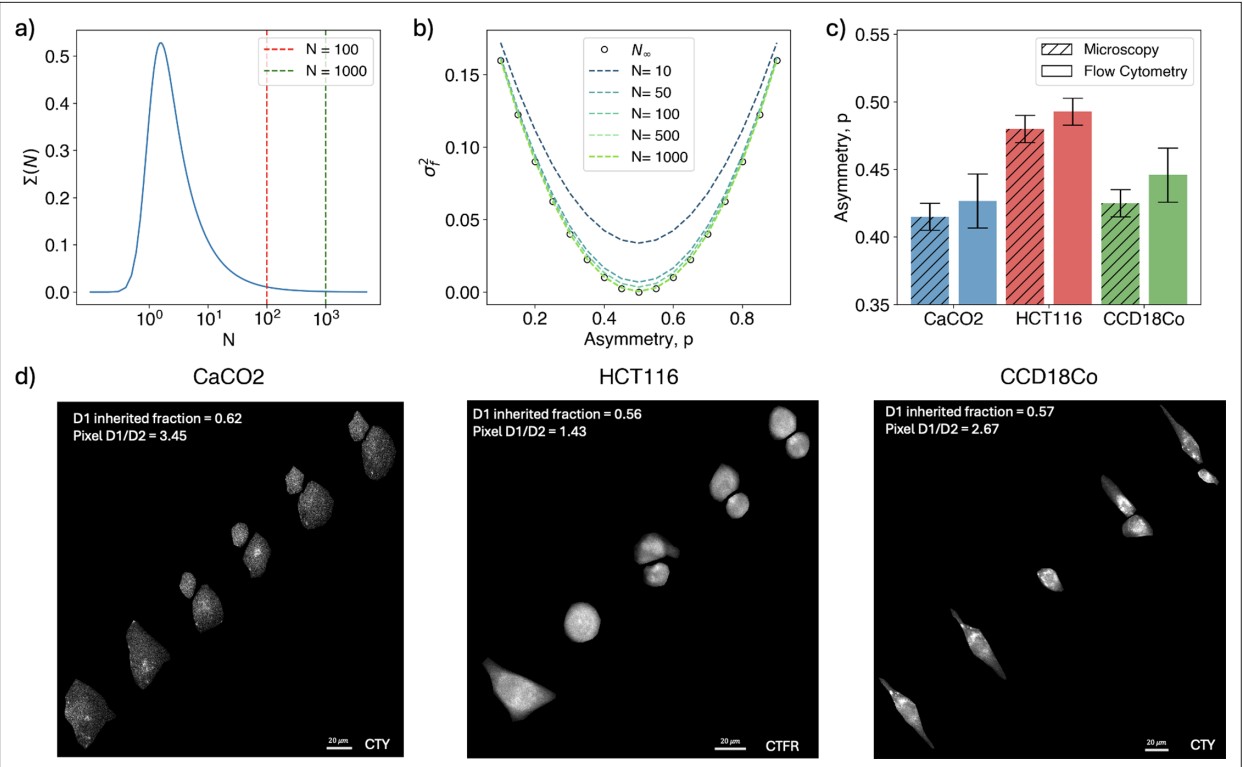

**Figure 4.** Cytoplasm partition fluctuations versus cell size. (**a**) Behavior of the integral term $\Sigma(N)$ (**Equation 17**) as a function of the number of dividing elements, $N$, at fixed $\sigma_{N_i} = 0.8$. Vertical lines mark typical values of cellular elements, like mitochondria. (**b**) Theoretical value of the variance of $\Pi(f)$ in the binomial assumption for different levels of asymmetries and increasing values of $N$. (**c**) Asymmetry of the partitioning distribution $\Pi(f)$ in the binomial limit as measured by the binomial bias, $p$. (**d**) Sample cases of volume asymmetric division for different cell lines. Images show the overlay of consecutive times during the division dynamic. Time flows from the bottom left to the top right.

$$\sigma_f^2 = \frac{1}{2}e^{\sigma_{N_i}^2/2-\mu}(1 - erf(\frac{\sigma_{N_i}^2 - \mu}{\sqrt{(2)}\sigma_{N_i}})) + \frac{p^2+q^2}{2} - \frac{1}{4}, \tag{18}$$

where $\mu = \langle \log(N_i) \rangle$.

The quantity $\sigma_{N_i}$ can be computed from flow cytometry or microscopy measurements. Indeed, if we consider the relation $I = N_i F$, where $I$ is the total fluorescence intensity, $N_i$ the number of components, and $F$ the fluorescence of each component, then $\sigma_I^2 = \sigma_{N_i}^2$. Since $\sigma_I^2$ is accessible from the data, so is $\sigma_{N_i}$. On the other hand, the values of $\mu$ are not directly accessible, limiting the possibility of an exact analytical computation. However, by computing the value of $\Sigma(N)$ as a function of $N$ for fixed values of $\sigma_{N_i}$ (**Figure 4a-b**), one can see that already for $N = 100$ the role of $\Sigma(N)$ is negligible. In general, for the partitioning of components such as cytoplasm, which are present in much larger numbers in the cell, this factor can be neglected.

In this regime, the expression for $\sigma_f^2$ becomes:

$$\sigma_f^2 = \frac{p^2+q^2}{2} - \frac{1}{4}. \tag{19}$$

By inverting this relationship, we obtained a general mapping between the variance and the level of biased asymmetry of the binomial distribution.

$$p = \frac{1}{2} - \sigma_f. \tag{20}$$

In **Figure 4c**, we validated the analytical expression against the values obtained via microscopy measurements, showing again good compatibility of the results.

Caco2 cells confirm to be the most asymmetric among the cell lines, and since they are known for their heterogeneity in cell morphology (*Lea, 2015*), we explored the hypothesis that shape heterogeneity and cytoplasmic asymmetry are linked. *Figure 4d* displays three sample cases of division examined through time-lapse fluorescent microscopy following cytoplasm. Different time frames are overlaid to depict the entire dynamic at once, with time progressing from the bottom left to the top right of each image.

We computed the pixel size ratio between the daughter cells in the last analyzed frame and compared it to the fraction of cytoplasm inherited from the mother cell. Although pixel size is only a rough proxy for cell volume, we observe that the cell inheriting a larger fraction of cytoplasm also appears larger in size. This finding suggests that asymmetry in cytoplasmic partitioning is linked to size fluctuations at birth, where cell size reflects the bias observed in the segregation process.

## Discussion

Cell division is orchestrated by hundreds of molecular interactions, which are intrinsically stochastic processes (*Bialek, 2012*). The presence of such stochasticity results in the insurgence of variability among the cell phenotypes sharing the same genome, as is the case for cancer cells. In this respect, gene expression noise has been extensively characterized (*Elowitz et al., 2002*), incorporating cell cycle effects (*Thomas et al., 2018*; *Swain et al., 2002*; *Chen et al., 2004*), as well as the different strategies that cells evolved to regulate the extent of the noise in these channels (*Osella et al., 2011*; *Miotto, 2019*).

Besides such noise sources, proliferating cells are subject to the fluctuations originating from the partition of cellular elements at division. Analysis of this partition noise highlighted that under some circumstances, its contribution exceeds the others in creating heterogeneity (*Huh and Paulsson, 2011*; *Soltani et al., 2016*). In fact, asymmetrically dividing cells can produce daughters that differ in size, cellular components, and, in turn, fate (*Chhabra and Booth, 2021*).

In this framework, the origin of such asymmetry may be ascribed to the basal stochasticity of molecular processes taking place at different levels during the division and/or to specific, evolutionarily conserved mechanisms used by cells to control cell fate and generate diversity (*Sunchu and Cabernard, 2020*; *Miotto and Monacelli, 2020*). In fact, asymmetric partitioning of components alters the initial counts of molecules in the following cell cycle, which can lead the system to a different phenotypic state (*Moris et al., 2016*; *Miotto et al., 2023*). Here, we show that an accurate determination of partition fluctuations can be obtained via standard flow cytometry measurements of properly marked cell populations. Our approach is easier and faster with respect to the use of microscopy imaging, which requires the acquisition and analysis of extensive recordings of the cells, ultimately limiting the reachable statistics. In addition, it is specifically designed to isolate partitioning noise from other sources via the usage of non-endogenous dyes to target cell components.

Although the need to use live fluorescent dyes to track proliferation surely limits the extent of applicability of our method to markable components, we anticipate that there are numerous contexts in which our approach could give a determining insight. For instance, organelles such as mitochondria, the endoplasmic reticulum, lysosomes, peroxisomes, and centrosomes can be easily marked (*Peruzzi et al., 2021*; *Miotto et al., 2025*) and their unequal distribution between daughter cells has been observed to provoke functional differences that influence their fate. In this respect, asymmetric division of mitochondria in stem cells is associated with the retention of stemness traits in one daughter cell and differentiation in the other (*Katajisto et al., 2015*) and, although the functional consequences are not yet fully determined, with therapeutic resistance in cancer stem cells (*Hitomi et al., 2021*; *García-Heredia and Carnero, 2020*). Asymmetric inheritance of lysosomes, together with mitochondria, appears to play a crucial role in determining the fate of human blood stem cells (*Loeffler et al., 2022*). Furthermore, our approach could be extended to investigate cell growth dynamics and the origins of cell size homeostasis in adherent cells (*Miotto et al., 2023*; *Miotto et al., 2024*; *Kussell and Leibler, 2005*; *McGranahan and Swanton, 2017*; *De Martino et al., 2019*).

Here, we measured the segregation statistics of the cytoplasm of different colon cell types, finding that they are neatly distinct. Caco2 cells exhibit the highest degree of cytoplasmic partition fluctuations, while HCT116 division is the one with the lowest asymmetry. It is interesting to note that normal colon fibroblasts sit in between, suggesting that the cancer-associated deregulation of the normal cell activities leads to the reactivation of asymmetric cell division in a cancer-specific manner.

Indeed, Caco2 cells are known to display a markedly heterogeneous distribution of phenotypes (**Lea, 2015**). Notably, comparing the degree of asymmetry in the division of the cytoplasm in the three different analyzed cell types, we found that microscopy-derived fluctuations are systematically higher than those measured by flow cytometry. While this could be linked to the different protocols used, it seems more probable that the difference is due to a systematic overestimation of the fluctuations measured by microscopy. Indeed, we identified at least two possible origins of such bias: fluctuations in the fluorescence intensity measures and errors in the segmentation protocol. Regarding the first one, we demonstrated its effect through simple simulations that mimicked the microscopy analysis. While for the second, relying on manual segmentation not only reduces the data but might also introduce biased cell identification errors. Both these effects become important when data filtering is only partially allowed due to the low available statistics. A discussion on this topic is proposed in the Appendix.

We have assessed the stability of our model with respect to perturbations arising from the co-occurrence of multiple noise sources. In particular, we explored the case where fluctuations at division and variability in cell cycle length are dependent. In particular, we investigated a strongly coupled scenario in which division is triggered upon reaching a specific size (sizer strategy) and partitioning fluctuations correspond to size inheritance. We observed that—even under this extreme condition—our approach yields reliable parameter estimates that remain consistent with the experimental error. Moreover, we observed coupling-dependent increase in $\mu_g$ for newer generations. This observation opens new questions and potential applications for our method.

Finally, to get insights into the possible mechanisms behind the observed asymmetries, we looked for a specific shape of the partition distribution. Following a maximum entropy approach (**Bialek, 2012**; **Miotto and Monacelli, 2018**; **Miotto and Monacelli, 2021**), we started by considering the distribution that employs the fewest assumptions about the system, that is, we consider independent segregation, which directly maps into binomial partitioning. Indeed, this is a common assumption in segregation models that traces back to the pioneering works of **Berg, 1978**; **Rigney, 1979**. Notably, our results indicated biased segregations with colon cancer cells displaying different levels of biases. To explain their origin, we looked at the relative sizes of the daughter cells right after division. The found correlation between daughter sizes and the fractions of inherited fluorescence vouches size as the origin of the observed biased segregation.

In conclusion, by combining experimental data with statistical modeling, we show how it is possible to use flow cytometry data to extract reliable estimates of the strength of fluctuations during cell division. Our approach has the potential to be applied to different cell types, where a quantification of the level of division asymmetries that daughter cells can experience may provide new insights into the mechanisms of asymmetric cell division and its role in cancer heterogeneity and plasticity.

## Materials and methods

### Cell culture

Cell lines were purchased from ATCC and were routinely checked for mycoplasma contamination. Caco2, colorectal adenocarcinoma cell line, was maintained in complete culture media DMEM (D6046) containing 20% FBS, penicillin/streptomycin plus glutamine, nonessential amino acids (NEAA), and sodium pyruvate, all 1/100 dilution. HCT116 VIM RFP, colorectal carcinoma cell line, was maintained in complete culture media McCoy's 5A (M9309) containing 10% FBS, penicillin/streptomycin, plus glutamine.

CCD18Co, colon fibroblast cell line, was maintained in complete culture media DMEM containing 10% FBS, penicillin/streptomycin plus glutamine, NEAA, and sodium pyruvate, all 1/100 dilution.

All cells were kept in culture at 37°C in 5% $CO_2$ and passaged according to the experimental protocol. For all experiments, cells were harvested, counted, and washed twice in serum-free solutions and resuspended in room temperature PBS w/o salts for further staining.

In detail, to detach cells from culture flasks, we used Trypsin EDTA 0.25%. Culture media was removed and the flasks were washed once with PBS, then Trypsin EDTA was added and allowed to work for 1 min in an incubator at 37°C. Once the cells appeared detached, the flasks were mechanically agitated to facilitate the cells' loss of adhesion. Trypsin was then inactivated with one volume of complete media, and cells were collected, pelleted, and washed a second time in PBS. To determine

cell viability, prior to dye staining, the collected cells were counted with the hemocytometer using Trypan Blue, an impermeable dye not taken up by viable cells.

## Flow cytometry and cell sorting

To track cell proliferation by dye dilution for establishing the progeny of a mother cell, cells were stained with CellTrace Violet (CTV). The CTV dye staining (C34557, Life Technologies, Paisley, UK), used to monitor multiple cell generations, was performed according to the manufacturer's instructions, diluting the CTV 1/1000 in 0.5–1 ml of PBS for 20 min in a water bath at 37°C, mixing every 10 min. Afterward, 5x complete media was added to the cell suspension for an additional 5 min incubation before the final washing in PBS.

Labeled cells were sorted using a FACSAriaIII (Becton Dickinson, BD Biosciences, USA) equipped with Near UV 375, 488, 561, and 633 nm lasers and FACSDiva software (BD Biosciences version 6.1.3). Data were analyzed using FlowJo software (Tree Star, version 10.7.1). Briefly, cells were first gated on single cells, by doublet exclusion with morphology parameters area versus width (A vs. W), both side and forward scatter. The unstained sample was used to set the background fluorescence. The sorting gate was set around the maximum peak of fluorescence of the dye distribution. In this way, the collected cells were enriched for the highest fluorescence intensity. Following isolation, an aliquot of the sorted cells was analyzed with the same instrument to determine the post-sorting purity and population width, resulting in an enrichment of >97% for each sample. To monitor multiple cell divisions, the sorted cell population was seeded in 12-well plates (Corning, Kennebunk, ME, USA) at a cell density between $[30-70]\cdot10^3$ cells/well, according to the experiment, and kept in culture for up to 84 hr. Each well corresponds to a time point of the acquisition, and cells in culture were analyzed every 24, 36, 48, 60, 72, and 84 hr by the LSRFortessa flow cytometer. In order to set the time zero of the kinetic, prior to culturing, a tiny aliquot of the collected cells was analyzed immediately after sorting at the flow cytometer. The unstained sample was used to set the background fluorescence as described above.

## Time-lapse microscopy

To better investigate the cell proliferation dynamics, we performed time-lapse experiments for up to 3 days. In our previous paper (*Peruzzi et al., 2021*), we verified that cell growth is not affected by different dye combinations. Therefore, in the present work, we used both CellTrace Yellow (C34573 A, Life Technologies, Paisley, UK) and CellTrace Far Red (C34572, Life Technologies, Paisley, UK) to stain the cytoplasm for microscopy analysis. Low passage cells at around 60% confluency were counted and stained with CellTrace dyes 1/500 and plated on IBIDI cell imaging chambers (μ-Slide 4 and 8 wells) at a low cell density of $10^3$ cells/well. After overnight incubation, the chamber is transferred to the inverted microscope adapted with an incubator to keep cells in appropriate growing conditions. Brightfield and fluorescent confocal image stacks were acquired with a 20x air objective (Olympus, Shinjuku, Japan) and Zen Microscopy Software (Zeiss, Oberkochen, Germany), every 20 min.

Time-lapse images were analyzed using ImageJ and in-house Python programs.

## Acknowledgements

This research was partially funded by grants from ERC-2019-Synergy Grant (ASTRA, No. 855923); EIC-2022-PathfinderOpen (ivBM-4PAP, No. 101098989); Project 'National Center for Gene Therapy and Drugs based on RNA Technology' (CN00000041) financed by NextGeneration EU PNRR MUR—M4C2—Action 1.4—Call 'Potenziamento strutture di ricerca e creazione di campioni nazionali di R&S' (CUP J33C22001130001). MM acknowledges the contribution of the Italian Ministero dell'Università e della Ricerca, Decreto Ministeriale No. 1236 of August 1, 2023—FIS 2 CALL. Project FIS2023-02957 (CUP B53C24009530001), 'Fathoming oUt the role of partitioninG noIse in cancer epithelial-mesenchymal Transitions (FUGIT).

## Additional information

### Funding

| Funder | Grant reference number | Author |
| --- | --- | --- |
| European Research Council | 855923 | Giancarlo Ruocco |
| European Innovation Council and Small and Medium-sized Enterprises Executive Agency | 101098989 | Giancarlo Ruocco |
| Italian Ministry of University and Research | B53C24009530001 | Mattia Miotto |
| National Center for Gene Therapy and Drugs based on RNA Technology financed by EU PNRR MUR | J33C22001130001 | Giancarlo Ruocco |

The funders had no role in study design, data collection, and interpretation, or the decision to submit the work for publication.

### Author contributions

Domenico Caudo, Data curation, Formal analysis, Investigation, Visualization, Writing – original draft, Writing – review and editing; Chiara Giannattasio, Simone Scalise, Data curation, Investigation; Valeria de Turris, Resources, Investigation; Fabio Giavazzi, Resources, Supervision; Giancarlo Ruocco, Resources, Supervision, Funding acquisition; Giorgio Gosti, Conceptualization, Validation, Investigation, Methodology; Giovanna Peruzzi, Data curation, Formal analysis, Supervision, Investigation, Methodology; Mattia Miotto, Conceptualization, Formal analysis, Supervision, Funding acquisition, Validation, Investigation, Methodology, Writing – original draft, Project administration, Writing – review and editing

### Author ORCIDs

Domenico Caudo (iD) https://orcid.org/0009-0002-1523-1939
Valeria de Turris (iD) https://orcid.org/0000-0003-0872-185X
Giorgio Gosti (iD) https://orcid.org/0000-0002-8571-1404
Mattia Miotto (iD) https://orcid.org/0000-0002-0043-8921

Reviewer #1 (Public review): https://doi.org/10.7554/eLife.104528.4.sa1
Reviewer #2 (Public review): https://doi.org/10.7554/eLife.104528.4.sa2
Author response https://doi.org/10.7554/eLife.104528.4.sa3

## Additional files

### Supplementary files

MDAR checklist

Source data 1. Data to reproduce the results shown in the Figures.

### Data availability

All relevant data are within the Main Text and the Supplementary Material. All codes used to produce the findings of this study are available at https://github.com/ggosti/fcGMM (copy archived at *Gosti, 2024*).

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

## Appendix 1

### A. General model for the partitioning noise

Recalling the computation made in previous work (*Peruzzi et al., 2021*), we want to compute an analytical formula for the expected mean and variance of the distribution of components in a proliferating population.

We assume that the initial marked compounds follow a Gaussian distribution with mean $\mu$ and variance $\sigma^2$. This assumption is experimentally forced by the gating strategy. Due to proliferation, the compounds will be successively divided by the daughter cells, and their distribution will be affected by the properties of the partitioning process.

We will indicate with $m_i$ the number of compounds in the mother cell and with $m_{2i}$ and $m_{2i+1}$ the compounds in the two daughter cells. The probability of finding a cell with $m_{2i}$ marked compounds at generation $g$ is:

$$P(m_{2i}|m_i) = \int \delta(m_{2i} - fm_i)\Pi(f)df,$$
$$P(m_{2i+1}|m_i) = \int \delta(m_{2i} - (1-f)m_i)\Pi(f)df'$$

$$(21)$$

where $\Pi(f)$ is the distribution probability of the fraction $f$ of compound inherited by the daughter cells. We will omit the computation for $m_{2i+1}$ because it is equal to $m_{2i}$, but to substitute $f$ with $1-f$. So:

$$P(m_{2i}) = \int P(m_{2i}|m_i)P(m_i)dm_i$$

which, according to *Equation 21*, becomes:

$$P(m_{2i}) = \int dfdm_i\delta(m_{2i} - fm_i)\Pi(f)P(m_i).$$

$$(22)$$

Hence, the expectation value for $m_{2i}$ is:

$$\mu_{2i} = E[m_{2i}] = \int dfdm_iP(m_{2i})m_{2i}dm_{2i}$$
$$= \int dfdm_idm_{2i}m_{2i}\delta(m_{2i} - fm_i)\Pi(f)P(m_i)$$
$$= E[f]E[m_i] = \mu_f\mu_i,$$

$$(23)$$

while for $m_{2i+1}$:

$$\mu_{2i+1} = E[m_{2i+1}] = \int dfdm_iP(m_{2i+1})m_{2i+1}dm_{2i1+}$$
$$= \int dfdm_idm_{2i+1}m_{2i+1}\delta(m_{2i+1} - (1-f)m_i)\Pi(f)P(m_i)$$
$$= E[1-f]E[m_i] = \mu_{1-f}\mu_i.$$

For the variance, we can do a similar computation:

$$\sigma_{2i} = E[m_{2i}^2] - E[m_{2i}]^2$$
$$= \int dfdm_idm_{2i}m_{2i}^2\delta(m_{2i} - fm_i)\Pi(f)P(m_i) - \mu_f^2\mu_i^2$$
$$= E[f^2]E[m_i^2] - \mu_f^2\mu_i^2 = E[f^2]\sigma_i^2 + \mu_i^2\sigma_f^2.$$

$$(24)$$

Therefore, both variance and mean of the daughters subpopulations are linked to the variance and mean of the mother's and to the distribution probability of $f$. To be able to fit the model with experimental data, we need to compute the mean and variance of the whole population, which is the sum of all coexisting generations. So the mean of the component distribution at generation $g$ will be:

$$\mu_g = \frac{1}{2^g}\sum_{k=1}^{2^g}\mu_g^k.$$

$$(25)$$

Here, we are assuming that a mother cell always gives rise to two daughters. We can split *Equation 25* into two groups, the one that always inherits a fraction $E[f]$ of the mother's compounds and the symmetrical one which inherits a fraction $E[1-f]$.

$$\mu_g = \frac{1}{2^g} \left[ \sum_{k=1}^{2^{g-1}} E[f]\mu_{g-1}^k + \sum_{k=1}^{2^{g-1}} E[1-f]\mu_{g-1}^k \right].$$

That can be rewritten as:

$$\mu_g = \frac{1}{2^g} \sum_{k=1}^{2^{g-1}} \mu_{g-1} = \frac{1}{2}\mu_{g-1}. \tag{26}$$

Therefore, knowing the initial mean value of the population, $\mu_0$ we have:

$$\mu_g = \left(\frac{1}{2}\right)^g \mu_0. \tag{27}$$

*Equation 27* shows that the mean value of the number of compounds for each generation is independent of the underlying process and does not depend on the distribution $\Pi(f)$. Although expected, it does not allow us to infer anything from the experiments.

To obtain a dependence on the partitioning process, we need to compute the second moment. It can be demonstrated that the total variance of the mixture of multiple distributions can be written as follows:

$$\sigma_g^2 = \frac{1}{2^g} \sum_{k=1}^{2^g} \left( \sigma_{g,k}^2 + \mu_g^2 \right) - \mu_g^2. \tag{28}$$

As we did in *Equation 26*, we can rewrite *Equation 28* by splitting into the two subpopulations of daughters:

$$A_g \quad = \frac{1}{2^g} \sum_{k=1}^{2^g} \left( \sigma_{g,k}^2 + \mu_{g,k}^2 \right)$$

$$= \frac{1}{2^g} \left[ \sum_{k=1}^{2^{g-1}} \left( \sigma_{g,k}^{2(f)} + \mu_{g,k}^{2(f)} \right) + \sum_{k=1}^{2^{g-1}} \left( \sigma_{g,k}^{2(1-f)} + \mu_{g,k}^{2(1-f)} \right) \right].$$

We can rewrite this formula recalling the relationships *Equations 23 and 24*.

$$A_g = \frac{1}{2^g} \sum_{k=1}^{2^g} \left( \sigma_{g,k}^2 + \mu_g^2 \right)$$

$$= \frac{1}{2^g} \left[ \sum_{k=1}^{2^{g-1}} E[f^2]\sigma_{g-1,k}^2 + \sigma_f^2 \mu_{g-1,k}^2 + \mu_f^2 \mu_{g-1,k}^2 \right. \tag{29}$$

$$\left. + \sum_{k=1}^{2^{g-1}} E[(1-f)^2]\sigma_{g-1,k}^2 + \sigma_{1-f}^2 \mu_{g-1,k}^2 + \mu_{1-f}^2 \mu_{g-1,k}^2 \right]$$

The last two elements of every sum can be rewritten as follows:

$$\sigma_f^2 \mu_{g-1,k}^2 + \mu_f^2 \mu_{g-1,k}^2 = E[f^2]\mu_{g-1,k}^2,$$

$$\sigma_{1-f}^2 \mu_{g-1,k}^2 + \mu_{1-f}^2 \mu_{g-1,k}^2 = E[(1-f)^2]\mu_{g-1,k}^2$$

Leading to:

$$A_g = \frac{1}{2^g} \left( \sum_{g-1,k}^{2} \sigma_{g-1,k}^2 + \mu_{g-1,k}^2 \right) \left( E[f^2] - E[(1-f)^2] \right)$$

$$= \frac{1}{2} A_{g-1}(1 - 2\mu_f + 2E[f^2]),$$

and considering $\mu_f = 1/2$.

$$A_g = A_{g-1}E[f^2]. \tag{30}$$

Therefore, the final form of *Equation 26* is:

$$\sigma_g^2 = A_g - \mu_g^2 = A_{g-1}E[f^2] - \mu_f^2 \mu_{g-1}^2$$

$$= A_0 E[f^2]^g - \mu_f^{2g} \mu_0^2 = \mu_0^2 (E[f^2]^g - \mu_f^{2g}) + \sigma_0^2 E[f^2]^g \tag{31}$$

$$= \mu_0^2 (E[f^2]^g - (1/2)^{2g}) + \sigma_0^2 E[f^2]^g.$$

## B. Model's stability to concurrence of noise sources

In the General model formulation (Section A), we have made the strong assumption of neglecting all the noise sources affecting component's count, but for the partitioning noise. In this section, we want to clarify the reasons for this assumption and evaluate the stability of our method in the presence and co-occurrence of other noises.

A cell proliferation cycle is affected by different sources of variability: (1) production and degradation processes of molecules; (2) variability in length of the cell cycle; and (3) partitioning noise, which identifies asymmetric inheritance of components between the two daughter cells. However, the experimental approach and the model have been formulated so as to specifically address the effects of partitioning noise. Indeed, since we are dealing with components tagged via live fluorescent markers, production of new fluorophores is impossible and can therefore be discarded. Instead, the degradation process is a global effect that influences the behavior of the mean of the distribution in a time-dependent manner. However, by looking at the experimental data in *Figure 1* of the main text, no significant depletion of fluorescence is observed, or at least it is hidden by the experimental fluctuations of the measure. Instead, a more careful evaluation has to be done for what concerns fluctuation in cell cycle length. We reasoned that in the assumption of independence between the noise sources, variability in cell cycle length would affect the timing of population emergence but not the intrinsic properties of those populations (e.g., Gaussian variance). To test this hypothesis, we conducted two sets of simulations. In the first, we assumed the aforementioned independence between fluctuations in cell cycle length and partitioning noise. We modeled the cells' proliferation with an event-driven algorithm (*Gillespie, 1977*). To each cell is associated a division time drawn from an Erlang distribution (mean = 18 hr, $k = 4$) and a number that corresponds to the stained component at time 0, which is extracted from a Gaussian distribution ($\mathcal{N}(\mu_0, \sigma_0)$). The only possible event is cell division, and the events are ordered by happening time in a heap tree structure. At every division event, two daughter cells are created; they inherit a fraction $f$ and $1 - f$ of their mother's components and are inserted in the population while their mother is removed. The value of $f$ is extracted from a Gaussian distribution ($\mathcal{N}(p, \sigma_p)$). The results, showing the behavior of the mean and variance of the component distributions across generations, are presented in *Figure 1—figure supplement 1*. Under the assumption of independence between these different noise sources, no significant effects were observed, and we are able to perfectly reconstruct the dynamic with our theoretical model. The second set of simulations considered a situation in which the cell's components and division time are coupled. Indeed, cells may adopt different growth and division strategies, which can be grouped into three categories based on what triggers division:

- Sizer-like cells divide upon reaching a certain size.
- Timer-like cells divide after a fixed time (corresponding to the previously treated case with independent noise).
- Adder-like cells divide once their volume has increased by a finite amount.

A detailed discussion of these strategies, including their mathematical formulation, can be found in *Miotto et al., 2024*, to which we refer the reader for further information. Here, we will assume that cells follow a sizer-like model ($\alpha = 1, \beta = 6$). In this way, we study a system in which cells with a higher

number of components have shorter division times. Hence, older (newer) generations are emptied (populated) starting from higher values. The simulations were performed using stochastic simulations of the growth and dividing cell population. In particular, simulations were performed starting from $N = 10,000$ initial cells. Each cell has two properties: its size and the number of stained components it carries. The initial size of the cells is extracted from a Gaussian distribution ($\mathcal{N}(\mu_s, \sigma_s)$) which at time 0 is equal to the stained component distribution. For each cell, a division time is extracted from the probability distribution $P(t_d)$ via inverse transform sampling. For the considered system, $P(t_d)$ is given by **Nieto-Acuna et al., 2019** :

$$P(t_d) = 1 - \exp\left(-\int_0^{t_d} h(s)\,ds\right).$$

Upon division, each cell is split into two new daughter cells, each inheriting a fraction $f$ and $1 - f$ of the mother size and component, respectively, extracted from a Gaussian distribution $N(p, \sigma_p)$. The value of the other parameters chosen for the simulation is fixed to $\kappa = 0.08, \lambda = 1, \alpha = 1, \beta = 6, Q = 5, \mu_s = 1, \sigma_s = 0.01$, while $p$ and $\sigma_p$ are varied. The results of the simulated systems and a comparison with the expected dynamic for $\mu_g$ and $\sigma_g$ are shown in **Figure 1—figure supplement 2**. As can be observed, higher levels of division asymmetry increase the fluctuations of the system relative to the analytically expected behavior, particularly in later generations. As previously noted, the distribution of components for a given generation is both populated and depleted from the top, which affects both the variance and the mean. This explains the greater discrepancy between expected and observed measurements in higher generations. Indeed, the population has not had sufficient time to fully proliferate, resulting in an incomplete sampling and thus a biased estimate of the distribution's properties.

Therefore, we assessed the accuracy of the method in reconstructing the strength of fluctuations ($\sigma_f$) by fitting the simulation results with the analytical expression in **Equation 31** and comparing them with the true values. The results are shown in **Figure 1—figure supplement 3**. As expected, the error in reconstructing the variance of the partitioning distribution increases with the level of asymmetry, reaching up to 20% for the most asymmetric case. This indicates that noise coupling has a significant effect on the uncertainty of our methodology. However, even under the extreme assumption of maximum coupling between partitioning noise and division time, the resulting uncertainty remains within the range of experimental noise (see **Figure 3d** in the main text). Moreover, due to the direction of the bias introduced by the coupling, the method can only underestimate the true asymmetry, making our estimate a conservative lower bound. In conclusion, while noise coupling does impact the measurement, its effect remains well within the error margins already accounted for.

## C. Modeling partition as a binomial process

The model we have developed enables a robust characterization of partitioning noise, as confirmed by two independent sets of experiments. By exploiting the high-throughput capabilities of flow cytometry, we can reliably measure the variance of the partitioning distribution by tracking the proliferation dynamics of an entire cell population. However, the general framework does not allow for a complete reconstruction of the partitioning distribution. A degeneracy remains in identifying the source of asymmetry. Indeed, by measuring only the variance, we gain no information about whether the process is biased. The distribution could be broad and symmetric, or narrow and asymmetric. HCT-116 and Caco2 may represent respective examples of these scenarios. In this section, we aim to resolve this degeneracy by introducing an additional assumption about the shape of the partitioning distribution.

We begin by recalling that the dyes used bind non-specifically to cytoplasmic amines. As a result, (1) the fluorescence is expected to be uniformly distributed throughout the cytoplasmic space, and (2) the number of labeled cytoplasmic components can be considered large. Within this framework, the simplest model one can adopt is a binomial distribution, with the parameter $p$ capturing the potential bias in the partitioning process. The binomial model assumes that each cellular component has a probability $p$ of being inherited by one of the two daughter cells, and a probability $q = 1 - p$ of being inherited by the other.

Therefore, the total partition distribution can be written as:

$$\Pi(f) = \frac{1}{2}(\Pi(f)^{(p)} + \Pi(f)^{(q)}), \tag{32}$$

where $f$ is the inherited fraction of components. As we have already observed, due to symmetry $\langle f \rangle = 1/2$. However, our interest lies in understanding how the general noise translates into the binomial model.

The variance of a single branch $\sigma_f^{2\,(p)}$, can be computed in the following way:

$$
\begin{aligned}
\sigma_f^{\,2\,(p)} &= \langle f^2 \rangle^{(p)} - \langle f \rangle^{2\,(p)} \\
&= \int df \int dN_i\, f^2\, \Pi(f \mid N_i)^{(p)} P(N_i) - \langle f \rangle^{2\,(p)} \\
&= \int dN_i \int df\, f^2\, \Pi(f \mid N_i)^{(p)} P(N_i) - \langle f \rangle^{2\,(p)} \\
&= \int dN_i \left( \sigma_{f|N_i}^{2\,(p)} + \langle f \rangle_{p|N_i}^2 \right) P(N_i) - \langle f \rangle_p^2 \\
&= \int dN_i\, \sigma_{f|N_i}^{2\,(p)} P(N_i),
\end{aligned}
\tag{33}
$$

where the notation $\sigma_{f|N_i}^{2\ (p)}$ indicates the variance of the distribution $p$ as a function of $f$ given a certain $N_i$. Consider, for example, the distribution of one of the two daughter cells, $\Pi(f)^{(p)}$. The average value of $f$ will be given by:

$$
\begin{aligned}
\langle f \rangle^{(p)} &= \int df \int dN_i f \Pi(f|N_i)^{(p)} P(N_i) \\
&= \int dN_i\, p\, P(N_i) = p,
\end{aligned}
$$

and similarly: $\langle f \rangle^{(q)} = q$. Recalling that $f$ is defined as $f = \frac{m_{2i}}{N_i}$, it follows that:

$$
\begin{aligned}
\sigma_{f|N_i}^{2\ (p)} &= \langle f^2 \rangle_{N_i}^{(p)} - \langle f \rangle_{N_i}^{2\ (p)} = \frac{1}{N_i^2}\sigma_x^{2(p)} \\
&= \frac{1}{N_i^2} N_i pq = \frac{pq}{N_i}.
\end{aligned}
\tag{34}
$$

Plugging *Equation 34* into *Equation 33* we get:

$$\sigma_f^{2(p)} = pq \int \frac{1}{N_i} P(N_i), \tag{35}$$

which in the case of a delta distribution (i.e., for a single value of $N_i$), consistently results in $\sigma^{2(p)} = \frac{pq}{N_i}$.

Once the main quantities of the individual distribution have been determined, we consider the distribution of the sum (*Equation 32*).

The mean value is:

$$\langle f \rangle = \int f P(f) df = \frac{1}{2}(\langle f \rangle^{(p)} + \langle f \rangle^{(q)}) = \frac{1}{2}(p + q) = \frac{1}{2}. \tag{36}$$

Therefore, to determine $p$, it is necessary to estimate the variance $\sigma_f^2$, which by definition is:

$$\sigma_f^2 = \langle f^2 \rangle - \langle f \rangle^2. \tag{37}$$

Substituting *Equation 36* one gets:

$$\sigma_f^2 = \frac{1}{2}(\langle f^2 \rangle^{(p)} - \langle f^2 \rangle^{(q)}) - \frac{1}{4}, \tag{38}$$

from which, adding and subtracting $\langle f \rangle^{2(p)}, \langle f \rangle^{2(q)}$, one gets:

$$
\begin{aligned}
\sigma_f^{\,2} &= \frac{1}{2} \left( \langle f^2 \rangle^{(p)} - \langle f \rangle^{2\,(p)} \right. \\
&\quad \left. + \langle f^2 \rangle^{(q)} - \langle f \rangle^{2\,(q)} + \langle f \rangle^{2\,(q)} \right) \\
&= \frac{1}{2} \left( \sigma_f^{\,2\,(p)} + p^2 + \sigma_f^{\,2\,(q)} + q^2 \right) - \frac{1}{4},
\end{aligned}
\tag{39}
$$

and via *Equation 35*

$$\sigma_f{}^2 = pq \int dN_i \frac{1}{N_i} P(N_i) + \frac{p^2 + q^2}{2} - \frac{1}{4}. \tag{40}$$

## 1. Delta distribution of mother's components

In principle, $P(N_i)$ can have any functional form, even a very complex one, which is not known. Therefore, let us start with the simplest case: all mother cells possess the same number of components $N_0$, so the distribution $P(N_i)$ is a Dirac delta.

In this case, the theoretical variance (*Equation 42*) will be:

$$\sigma_f{}^2 = pq \int dN_i \frac{1}{N_i} \delta(N_i - N_0) + \frac{p^2 + q^2}{2} - \frac{1}{4}, \tag{41}$$

$$= \frac{pq}{N_0} + \frac{p^2 + q^2}{2} - \frac{1}{4}. \tag{42}$$

Since the values of the initial distribution have been considered randomly, each of them will have the same probability of being selected: $P(N_i) = \frac{1}{N_{max} - N_{min}}$. Consequently, the integral term in the variance expression (*Equation 42*) can be simplified as:

$$\int dN_i \frac{1}{N_i} P(N_i) = \frac{1}{N_{max} - N_{min}} \int dN_i \frac{1}{N_i}$$
$$= \frac{\ln(N_{max}) - \ln(N_{min})}{N_{max} - N_{min}}, \tag{43}$$

and one gets to the final expression:

$$\Sigma(N) = \sigma_f{}^2$$
$$= pq \frac{\ln(N_{max}) - \ln(N_{min})}{N_{max} - N_{min}} + \frac{p^2 + q^2}{2} - \frac{1}{4}. \tag{44}$$

Neglecting the integral term, the equation reduces to:

$$\sigma_f^2 = \frac{p^2 + q^2}{2} - \frac{1}{4}. \tag{45}$$

## 2. Lognormal distribution of mothers' components

In an even more realistic case, the distribution of the number of internal components of the mother cells will be given by a log-normal distribution:

$$P(N_i) = \frac{e^{\frac{-(ln(N_i) - \mu)^2}{2\sigma_{N_i}^2}}}{\sqrt{2\pi\sigma_{N_i}^2} N_i}, \tag{46}$$

where $\mu$ is the average value: $\mu = \langle log(N) \rangle$.

The integral term in this case is given by:

$$\Sigma(N) = \int dN_i \frac{1}{N_i} P(N_i)$$
$$= \frac{1}{\sqrt{2\pi\sigma_{N_i}^2}} \int dN_i \frac{1}{N_i^2} e^{\frac{-(ln(N_i) - \mu)^2}{2\sigma_{N_i}^2}}. \tag{47}$$

In general, the distribution of internal components of a cell is not known. However, in microscopy experiments, it is possible to determine fluorescence distributions, and we could assume a direct proportionality between the two variables: $I = N_i \cdot F$ is a constant value representing the fluorescence of an internal component, $I$ is the total fluorescence, and $N_i$ the number of components.

Now, considering the variance of the distribution as a function of the number of internal components:

$$
\begin{aligned}
\sigma_{N_i}^2 = \sigma_{I/F}^2 &= \left\langle \ln\left(\frac{I}{F}\right)^2 \right\rangle - \left\langle \ln\left(\frac{I}{F}\right) \right\rangle^2 \\
&= \left\langle (\ln(I) - \ln(F))^2 \right\rangle - \left\langle \ln(I) - \ln(F) \right\rangle^2 \\
&= \left\langle (\ln^2(I)) \right\rangle - \langle \ln(I) \rangle^2 = \sigma_I^2.
\end{aligned}
\tag{48}
$$

The variance of the distribution as a function of $N_i$ coincides with that of the fluorescence. Consequently, for a log-normal distribution, the contribution of the integral term to the variance, integrated over the interval $[1, \infty)$, will be:

$$
\begin{aligned}
\Sigma(N) &= \int_1^\infty dN_i \frac{1}{\sqrt{2\pi\sigma_{N_i}^2}} \frac{1}{N_i^2} e^{\frac{-(ln(N_i) - \mu)^2}{2\sigma_{N_i}^2}} \\
&= \frac{1}{2} e^{\sigma^2/2 - \mu}(1 - erf(\frac{\sigma^2 - \mu}{\sqrt{(2)}\sigma})),
\end{aligned}
\tag{49}
$$

where the variance coincides with that of the fluorescence distribution (*Equation 48*) and $\mu = \langle \log(N) \rangle$.

Substituting the integral term (*Equation 49*) into the complete expression for the variance (*Hitomi et al., 2021*) for a log-normal distribution of the mother cells, we obtain:

$$
\begin{aligned}
\sigma_f^2 &= pq \cdot \frac{1}{2} e^{\frac{\sigma^2}{2} - \langle log(N) \rangle} \left( 1 - erf\left( \frac{\sigma^2 - \langle log(N) \rangle}{\sqrt{2}\sigma} \right) \right) \\
&+ \frac{p^2 + q^2}{2} - \frac{1}{4}.
\end{aligned}
\tag{50}
$$

## D. Gating strategy

The gating strategy plays a fundamental role in our model. Indeed, the ability to distinguish different cell generations based on the fluorescence intensity distribution at a given time point in the time course strongly depends on the degree of overlap between these distributions. The greater the overlap, the less accurately the Gaussian mixture model (GMM) can separate them. This overlap is influenced by the coefficients of variation (CV) of both the partitioning distribution and the initial component distribution. Specifically, the fluorescence intensity distribution at time $t$ results from the convolution of the distribution at time $t - 1$ with the partitioning distribution function. Therefore, the balance between partitioning asymmetry and the width of the initial component distribution is crucial. As shown in *Figure 2—figure supplement 2*, we simulated the proliferation of a cell population and plotted the distribution of cell components at a specific time point, distinguishing different generations by varying shades of blue. Increasing the CV of either distribution reduces the ability to distinguish between generations. However, the variance of the initial distribution cannot be reduced arbitrarily. While selecting a narrower distribution facilitates better reconstruction of the generations, it also limits the number of cells available for the experiment. Consequently, for components exhibiting a high level of asymmetry, further narrowing of the initial distribution becomes experimentally impractical. In such cases, an approach previously tested on liquid tumors (*Peruzzi et al., 2021*) involves applying the GMM in two dimensions by co-staining another cellular component with lower division asymmetry.

## E. Noise overestimation with time-lapse fluorescent microscopy

In the main document, we argue that the limited statistics available from time-lapse fluorescence microscopy may explain its tendency to overestimate fluctuations in cell division when compared to

flow cytometry. In this section, we aim to provide some arguments regarding this hypothesis. We identified at least two phenomena that go in the direction of producing an overestimation and that affect microscopy and not flow cytometry measurements. The first regards the inherent fluctuations in the single-cell fluorescence intensity measures. This variability adds to the inherent stochasticity of the partitioning process, thereby increasing the overall variance of the distribution. To illustrate this effect, we simulated variability in microscopy data and consecutively evaluated the effect in the inherited fraction estimation. We extracted a fraction $f$ from a Gaussian distribution with mean $\mu = p$ and standard deviation $\sigma = \sigma_{true}$, that is, $\mathcal{N}(p, \sigma_{true})$. We then reproduced different time frames by adding noise drawn from a Gaussian distribution with mean $\mu = 0$ and standard deviation $\sigma = \sigma_{noise}$, that is, $\mathcal{N}(0, \sigma_{noise})$, to $f$. The same procedure was applied to $1 - f$. The added noise was resampled to ensure that the two measurements remained independent. *Figure 3—figure supplement 1* shows an example dynamic, where the empty gray circles represent the true fractions.

Next, we fitted the two dynamics with linear equations, imposing a common slope, to obtain an estimate of the partitioning fraction. Repeating this process a number of times consistent with the experiment yielded the partitioning distribution, of which we measured the variance. *Figure 3—figure supplement 2* shows the distribution of the measured standard deviation over multiple repetitions of the entire simulation. Each histogram is the distribution of the obtained variances of the noisy and noise-free dynamics. By looking at the mean of the distribution, we can see how, on average, the estimated variance is greater than the true one. The magnitude of this increase depends on the properties of the added noise, but the bias is always toward larger values.

The distributions and simulations presented here are intended solely to demonstrate the direction of the bias, not to precisely account for the observed difference between flow cytometry and microscopy estimates in real data. In the case shown, where $\sigma_{true} = \sigma_{noise}$, the resulting difference in division asymmetry is 1.3.

A second contribution arises from the segmentation protocol. A major limitation of the microscopy-based approach is the need for manual image segmentation, which reduces the amount of usable data and introduces potential errors. Although several checks were applied, some issues are unavoidable. For example, when daughter cells are very close to one another, borders may not be clearly identified; cells may overlap; or speckles may be overlooked. In such cases, overestimation of fluorescence is more likely than underestimation, thereby increasing the probability of extreme events. Segmentation relies on both brightfield and fluorescence images. Hence, errors in defining cell outlines are more likely when fluorescence is low, since borders or overlaps become more difficult to distinguish if both cells are dark. This introduces an additional bias toward higher asymmetries, increasing the frequency of events in the tails of the partitioning distribution. Both sources of bias could be mitigated by increasing the available statistics. In particular, applying stricter selection criteria—such as setting limits on fluorescence intensity fluctuations—would help the measured distribution converge toward the expected one. By contrast, these issues do not arise in flow cytometry experiments. From the initial sorting procedure, which ensures a cleaner separation of peaks, to the morphological checks performed at each acquisition step, the availability of a large number of measured events reduces both measurement noise and segmentation errors.

