## [Editor Report · eLife Assessment]

This study presents a **useful** method based on flow cytometry to study partitioning noise during cell division. The methods, data and analysis support the claims of the authors is **convincing**. This work will be of interest to cell biologists and biophysicists working on asymmetric partitioning during cell division.

---

## [Referee Report · Reviewer #1 (Public review)]

Summary:

The aim of this paper is to develop a simple method to quantify fluctuations in the partitioning of cellular elements. In particular, they propose a flow-cytometry based method coupled with a simple mathematical theory as an alternative to conventional imaging-based approaches.

Strengths:

The approach they develop is simple to understand, and its use with flow-cytometry measurements is clearly explained. Understanding how the fluctuations in the cytoplasm partition varies for different kinds of cells is particularly interesting.

Weaknesses:

The theory only considers fluctuations due to cellular division events. Fluctuations in cellular components are largely affected by various intrinsic and extrinsic sources of noise and only under particular conditions does partitioning noise become the dominant source of noise. In the revised version of the manuscript, they argue that in their setup, noise due to production and degradation processes are negligible but noise due to extrinsic sources such as those stemming from cell-cycle length variability may still be important. To investigate the robustness of their modelling approach to such noise, they simulated cells following a sizer-like division strategy, a scenario that maximizes the coupling between fluctuations in cell-division time and partitioning noise. They find that estimates remain within the pre-established experimental error margin.

Comments on previous version:

The authors have addressed all of my comments.

---

## [Referee Report · Reviewer #2 (Public review)]

The authors present a combined experimental and theoretical workflow to study partitioning noise arising during cell division. Such quantifications usually require time-lapse experiments, which are limited in throughput. To bypass these limitations, the authors propose to use flow-cytometry measurements instead and analyse them using a theoretical model of partitioning noise. The problem considered by the authors is relevant and the idea to use statistical models in combination with flow cytometry to boost statistical power is elegant. The authors demonstrate their approach using experimental flow cytometry measurements and validate their results using time-lapse microscopy. The approach focuses on a particular case, where the dynamics of the labelled component depends predominantly on partitioning, while turnover of components is not taken into account. The description of the methods is significantly clearer than in the previous version of the manuscript.

---

## [Author Response]

The following is the authors’ response to the previous reviews

**Reviewer #1 (Public review):**
Summary:The aim of this paper is to develop a simple method to quantify fluctuations in the partitioning of cellular elements. In particular, they propose a flow-cytometry based method coupled with a simple mathematical theory as an alternative to conventional imaging-based approaches.Strengths:The approach they develop is simple to understand and its use with flow-cytometry measurements is clearly explained. Understanding how the fluctuations in the cytoplasm partition varies for different kinds of cells is particularly interesting.Weaknesses:The theory only considers fluctuations due to cellular division events. Fluctuations in cellular components are largely affected by various intrinsic and extrinsic sources of noise and only under particular conditions does partitioning noise become the dominant source of noise. In the revised version of the manuscript, they argue that in their setup, noise due to production and degradation processes are negligible but noise due to extrinsic sources such as those stemming from cell-cycle length variability may still be important. To investigate the robustness of their modelling approach to such noise, they simulated cells following a sizer-like division strategy, a scenario that maximizes the coupling between fluctuations in cell-division time and partitioning noise. They find that estimates remain within the pre-established experimental error margin.

We thank the Reviewer for her/his work in revising our manuscript.

**Reviewer #2 (Public review):**
Summary:The authors present a combined experimental and theoretical workflow to study partitioning noise arising during cell division. Such quantifications usually require time-lapse experiments, which are limited in throughput. To bypass these limitations, the authors propose to use flow-cytometry measurements instead and analyse them using a theoretical model of partitioning noise. The problem considered by the authors is relevant and the idea to use statistical models in combination with flow cytometry to boost statistical power is elegant. The authors demonstrate their approach using experimental flow cytometry measurements and validate their results using time-lapse microscopy. The approach focuses on a particular case, where the dynamics of the labelled component depends predominantly on partitioning, while turnover of components is not taken into account. The description of the methods is significantly clearer than in the previous version of the manuscript.

We thank the Reviewer for her/his work in revising our manuscript. In the following, we address the remaining raised points.

I have only two comments left:In eq. (1) the notation has been changed/corrected, but the text immediately after it still refers to the old notation.

We have fixed the notation.

Maybe I don't fully understand the reasoning provided by the authors, but it is still not entirely clear to me why microscopy-based estimates are expected to be larger. Fewer samples will increase the estimation uncertainty, but this can go either way in terms of the inferred variability.

We thank the Reviewer for giving us the opportunity to clarify this point. In the previous answer, we focused on the role of the gating strategy, highlighting how the limited statistics available with microscopy reduce the chances of a stronger selection of the events. The explanation for why the noise is biased toward increasing the estimation of division asymmetry relies on multiple aspects: First, due to the multiple sources of noise affecting fluorescence intensity, the experimental procedure, and the segmentation protocol, the measurements of the fluorescence intensity of single cells fluctuate. This variability adds to the inherent stochasticity of the partitioning process, thereby increasing the overall variance of the distribution.

To illustrate this effect, we simulated the microscopy data. We extracted a fraction f from a Gaussian distribution with mean µ = 𝑝 and standard deviation σ = σ_𝑡𝑟𝑢𝑒_ , i.e. 𝑁(𝑝, σ_𝑡𝑟𝑢𝑒_). We then simulated different time frames by adding noise drawn from a Gaussian distribution with mean µ = 0 and standard deviation σ = σ_𝑛𝑜𝑖𝑠𝑒_ , i.e., 𝑁(0, σ_𝑛𝑜𝑖𝑠𝑒_), to f. An equal process was applied to 1 − f. The added noise was resampled so that the two measurements remained independent. Figure 6 shows a sample dynamic where the empty gray circles represent the true fractions. We then fitted the two dynamics to a linear equation with a common slope and obtained an estimate of the partitioning noise.

By repeating this process a number of times consistent with the experiment, we measured the resulting standard deviation of the new partitioning distribution. Figure 7 shows the distribution of the measured standard deviation over multiple repetitions of the simulations. Each histogram is the variance of the partitioning distribution obtained from 100 simulations of the noisy (and non noisy) fluorescence dynamic. By comparing this with the distribution of the standard deviation of the non-noisy dynamics, it is possible to observe that, on average, the added noise leads to a greater estimated variance. The magnitude of this increase depends on the variance of the added noise, but it is always biased toward larger values.

This represents only one component of the effect. The shown distributions and simulations are intended solely to demonstrate the direction of the bias, and not to account for the exact difference between the flow cytometry and microscopy estimates. In the proposed case, where noise and true variance are equal, the resulting difference in division asymmetry is 1.3.

A second contribution arises from the segmentation protocol. As we stated, a major limitation of the microscopy-based approach is the need for manual image segmentation. This reduces the amount of available data and introduces potential errors. Even though different checks were applied, some situations are difficult to avoid. For example, when daughter cells are very close to each other, the borders may not be precisely recognized; cells may overlap; or speckles may remain undetected. In all these cases, it is easier to overestimate the fluorescence than to underestimate it, thereby increasing the chance of an extremal event.

Indeed, segmentation relies on both brightfield and fluorescence images. Errors in defining the cell outline are more likely when fluorescence is low, since borders, overlaps, and speckles are more evident against a darker background. This introduces an additional bias toward higher asymmetry, increasing the number of events in the tail of the partitioning distribution.

Both aspects described above could be mitigated by increasing the available statistics. In particular, by applying stricter selection criteria, such as imposing limits on fluorescence intensity fluctuations, the distribution should approach the expected one.

A similar issue does not arise in flow cytometry experiments. From the initial sorting procedure, which ensures a cleaner separation of peaks, to the morphological checks performed at each acquisition point, the availability of a large number of measured events reduces both measurement noise and segmentation errors.

A discussion on these aspects has been added in the revised version of the Supplementary Materials and in the Main Text.